# The Effects of Helmet Therapy Relative to the Size of the Anterior Fontanelle in Nonsynostotic Plagiocephaly: A Retrospective Study

**DOI:** 10.3390/jcm8111977

**Published:** 2019-11-14

**Authors:** Do Gon Kim, Joon Seok Lee, Jeong Woo Lee, Jung Dug Yang, Ho Yun Chung, Byung Chae Cho, Kang Young Choi

**Affiliations:** Department of Plastic and Reconstructive Surgery, School of Medicine, Kyungpook National University, Daegu 702701, Korea; fiter0726@knu.ac.kr (D.G.K.); leejspo@knu.ac.kr (J.S.L.); jeongwoo@knu.ac.kr (J.W.L.); lambyang@knu.ac.kr (J.D.Y.); hy-chung@knu.ac.kr (H.Y.C.); bccho@knu.ac.kr (B.C.C.)

**Keywords:** plagiocephaly, helmet therapy, anterior fontanelle, cranial vault asymmetry, posterior symmetry ratio, overall symmetry ratio

## Abstract

Helmet therapy is an important nonsurgical approach for patients with nonsynostotic plagiocephaly, but its effectiveness may depend on certain anatomical features. We retrospectively examined the effects of helmet therapy according to the size of the anterior fontanelle. Two hundred patients with nonsynostotic plagiocephaly who underwent helmet therapy between 2016 and 2018 were included. Data regarding age at treatment onset and treatment duration were collected. Patients were divided into two groups depending on the age at treatment initiation: the 12–23 weeks group and the >23 weeks group. Patients were also divided based on the anterior fontanelle size to analyze the effects of helmet therapy according to the severity of plagiocephaly in each group as the change in the cranial vault asymmetry index (CVAI). Therapeutic effects were evaluated using the cranial vault asymmetry (CVA), CVAI, anterior symmetry ratio, posterior symmetry ratio (PSR), and overall symmetry ratio at baseline and treatment completion. Treatment initiation at age 12–23 weeks resulted in better effects than that after age 24 weeks. There were no sex-dependent differences in therapeutic effects. Greater changes in the CVA, CVAI, and PSR were associated with larger anterior fontanelles. Therefore, the anterior fontanelle size could be a prognostic factor for estimating helmet therapy outcomes.

## 1. Introduction

Plagiocephaly refers to a skull with occipital flatness; the prevalence of nonsynostotic deformational or positional plagiocephaly, which occurs as a result of external factors such as molding, is higher than that of congenital plagiocephaly caused by internal factors such as craniosynostosis [1,2,3,4,5,6,7,8]. Deformational plagiocephaly is known to have diverse causes, which can be broadly divided into prenatal and postnatal. Prenatal causes include fetal position, multiple births, uterine compression, and intrauterine constraint, whereas postnatal causes include premature birth, assisted delivery for the first child, male sex, sleeping position, and torticollis. It is well known that the prevalence of positional plagiocephaly caused by sleeping position has substantially increased since the Back to Sleep campaign in the United States as an attempt to prevent sudden infant death syndrome (SIDS) [9].

The infant’s head varies in shape and grows quickly [10]. Therefore, continuously placing an infant in the supine position can easily induce positional plagiocephaly, thereby increasing its prevalence [11]. Furthermore, unilateral occipital plagiocephaly can cause contralateral frontal flatness; if this situation progresses, it may cause an ear shift, which leads to skull base and facial asymmetry [12,13,14]. Therefore, many studies have examined measures to prevent the progression of unilateral occipital plagiocephaly.

Of these measures, helmet therapy has been studied extensively since a 1981 report described its usefulness for patients with plagiocephaly [7]. Most of these studies reported that attending a craniofacial center and starting helmet therapy early (between ages 3 and 5 months) produce better outcomes compared to starting therapy after eight months of age [15].

We devised a cranial measurement method using the anteroposterior (AP) view of the skull and reported the usefulness of helmet therapy [10]. However, we discovered that there are varying effects, even among infants of the same age. While studying the causes, we hypothesized that the size of the anterior fontanelle may have a role in the effects because the anterior fontanelle increases in size during the first two months of life and then decreases after this period [10]. Therefore, it can be considered as an index of cranial growth in the prenatal period and the period after birth; consequently, it may impact the usefulness of helmet therapy.

We hypothesized that a larger anterior fontanelle size would lead to better outcomes of helmet therapy due to the easier cranial movement. Therefore, we conducted a retrospective analysis of the effects of helmet therapy according to the anterior fontanelle size.

## 2. Materials and Methods

This study was approved by the appropriate institutional review board (IRB 2019-02-026). We enrolled 200 patients with nonsynostotic plagiocephaly who underwent helmet therapy between 1 January 2016 and 31 December 2018. Helmet therapy was explained to the patients’ guardians if the cranial vault asymmetry (CVA) was 10 mm or more and the cranial vault asymmetry index (CVAI) was 3.5% or more. We recruited 200 infants with nonsynostotic plagiocephaly from 1 January 2016. The progress of the infants was observed using clinical photography after helmet therapy was conducted. We did not include preterm infants under 32 weeks in the study. Duc et al. reported that gestational age does not lead to any differences in the anterior fontanelle size after birth [16]. Therefore, we think that there were no differences due to gestational age, since all infants who underwent helmet therapy in our study were selected only after full term [16]. We divided patients into two groups at the start of treatment based on their age: a 12–23 weeks group and a ≥24 weeks group. Furthermore, we collected data regarding age at the start of treatment and treatment duration. We measured the anterior fontanelle size using X-ray imaging. The transverse length of the anterior fontanelle was measured in the skull AP view that was obtained to check craniosynostosis before treatment. We then divided the patients into three groups based on their anterior fontanelle size (34.3 mm ± 19.1 (SD)): group A, 0–25% (0–5.1 mm); group B, 25–75% (5.1–8.7 mm); group C, 75–100% (8.7 mm). In addition, 5-month-old infants were divided based on their anterior fontanelle size into group I (0–25%, 0–5.2 mm), group II (25–75%, 5.2–8.9 mm), and group III (75–100%, 8.9 mm). We first divided the patients into four groups in 25% intervals and combined the two middle groups to make a total of three groups. Based on the mean, the bottom half of the lower group was divided into group A, the top half of the upper group into group C, and the rest into group B. They were not divided into thirds because the values close to the mean in groups A and C would have obscured the statistical results. In addition, the percentages in the 3 groups (25%, 50%, and 25%) are easier to use clinically and to explain to caregivers.

Helmet treatment feasibility was evaluated using the cranial vault asymmetry (CVA), cranial vault asymmetry index (CVAI), anterior symmetry ratio (ASR), posterior symmetry ratio (PSR), and overall symmetry ratio (OSR) at baseline and at the end of treatment [17]. Two skull radiographs (anteroposterior and lateral) were examined to identify whether the cranial sutures were intact when plagiocephaly was clinically suspected, to exclude craniosynostosis. Then, the CVA and CVAI were measured using clinical photographs. Positional plagiocephaly was diagnosed when the CVA was >10 mm and the CVAI was >3.5%. Caregivers were then informed about helmet therapy. The distances between the orbitale superius, the supraorbital rim of the ipsilateral pupil, and the contralateral occiput were measured, and the absolute value of the difference between the two sides was defined as the CVA. The CVAI (normal: <3.5%; mild: 3.5–7%; moderate: 7–12%; severe: >12%) was computed by dividing the CVA by the smaller diagonal distance and multiplying it by 100 (Figure 1) [18].

Based on clinical photography using a digital camera (Canon EOS 750 D, Canon, Tokyo, Japan), the glabella, tragus, and opisthocranion were used as landmarks in the background with or without a grid at 5-mm intervals. Photography was performed in a specialized room in the same location for all patients. Each patient was placed in the supine position on a mayo table so that the anatomical landmarks were visible. Photographs were measured twice: first against a background without the grid and then against the background with the grid. The skull was divided into four quadrants using the lines connecting the glabella, tragus, and opisthocranion: Q1, anterior left quadrant; Q2, anterior right quadrant; Q3, posterior right quadrant; and Q4, posterior left quadrant. The symmetry ratios used for assessment were the ASR (Q1, Q2 smaller one/Q1, Q2 bigger one), PSR (Q3, Q4 smaller one/Q3, Q4 bigger one), and OSR (Q1 + Q4, Q2 + Q3) (Figure 2). The quadrant of each area was measured using NIH ImageJ software (National Institutes of Health and Laboratory for Optical and Computational Instrumentation, University of Wisconsin, Madison City, WI, USA), and symmetry was confirmed when each of the ratios was 0.9 or more (Figure 2).

## 3. Results

Two hundred patients participated in this study (105 males and 95 females). The mean treatment period was 13 weeks (range, 10–24 weeks). The mean total pre-helmet and post-helmet CVA values were 11.4 ± 1.5 mm and 5.36 ± 1.1 mm, respectively. Pre-helmet and post-helmet CVAI values were 12.73 ± 1.3% and 4.91 ± 0.8%. Regarding the patient distribution based on age at treatment initiation, 152 patients were in the 12–23 weeks group and 48 patients were in the ≥24 weeks group. The CVA values before and after treatment in the 12–23 weeks group were 11.4 and 7.36 mm, respectively, and the CVAI values were 8.73% and 5.4%, respectively, for a mean treatment duration of 12.5 weeks. In the ≥24 weeks group, the CVA values before and after treatment were 11.11 and 7.6 mm, respectively, and CVAI values were 8.5% and 5.5%, respectively, for a mean treatment duration of 16.3 weeks. No significant differences in treatment effects were seen between age groups (*p* = 0.345). Treatment duration was significantly longer (*p* < 0.05) in the ≥24 weeks group (Table 1).

Groups divided based on the anterior fontanelle size included 53, 102, and 45 infants in group A, group B, and group C, respectively (Table 2).

Differences in the CVA values between groups were 6.8 mm (A–B), 9.5 mm (A–C), and 2.7 mm (B–C); differences in the CVAI values were 7.28% (A–B; *p* < 0.05), 8.9% (A–C; *p* < 0.05), and 1.62% (B–C; *p* = 0.381) (Figure 3). There were no interval changes in the ASR in group A; for group B it changed 1%, from 0.9 to 0.91; and for group C it changed 2%, from 0.89 to 0.91. The PSR changed as follows: group A (6%), from 0.82 to 0.88; group B (10%), from 0.79 to 0.89; and group C (17%), from 0.73 to 0.9. In group A, the OSR changed 5%, from 0.85 to 0.90; in group B, it changed 9%, from 0.81 to 0.90; and in group C, it changed 13%, from 0.78 to 0.91 (Table 3, Figure 4).

Because of the possibility of an age-related bias, we additionally divided the 82 five-month-old infants (10–23 weeks) into three groups based on their anterior fontanelle size for a more accurate analysis of the treatment effects, with 21 patients in group I, 41 in group II, and 20 in group III. The CVA differences between groups were 8.2 mm (groups I and II), 10.1 mm (groups I and III), and 1.9 mm (groups II and III). The CVAI differences between groups were 8.41% (groups I and II; *p* < 0.05), 9.74% (groups I and III; *p* < 0.05), and 1.33% (groups II and III; *p* = 0.412). There were no interval changes in the ASR in group I; group II and group III changed 1%, from 0.9 to 0.91. For the PSR, group I changed 2%, from 0.84 to 0.86; group II changed 6%, from 0.82 to 0.88; and group III changed 13%, from 0.78 to 0.91. In group I, the OSR changed 3%, from 0.88 to 0.91; in group II, it changed 6%, from 0.86 to 0.92; and in group III, it changed 7%, from 0.86 to 0.93 (Table 4).

In addition, to verify the effectiveness of helmet therapy according to the severity of the plagiocephaly, the CVA values were divided into group X (*n* = 105), with a CVA value of less than 10, and group Y (*n* = 95), with a CVA value of ≥10 at the beginning of the treatment. In each group, the CVAI difference according to the anterior fontanelle size was measured before and after helmet treatment. To identify the correlation between the CVAI difference before and after helmet therapy according to the anterior fontanelle size, a linear regression analysis was conducted. Group X showed a positive correlation between the size of the anterior fontanelle and the CVAI difference that was statistically significant (*p* = 0.006). In addition, group Y showed a positive correlation between the size of the anterior fontanelle and the CVAI difference that was statistically significant (*p* = 0.003) (Table 5).

## 4. Discussion

Since plagiocephaly may cause facial asymmetry, we highlighted the importance of its assessment and treatment. To determine the effects of treatment, Silva et al. reported that the threshold for facial symmetry to detect maxillary dental midline shift is 2 mm [19]. However, Meyer-Marcotty et al. and Stellzig-Eisenhauer et al. reported that infants with a 2-mm difference between the nasion and forehead had normal symmetry [20,21].

First introduced by Clarren et al., helmet therapy has since been widely applied and extensively studied [7]. In 2004, Teichgraeber et al. reported that infants younger than six months and those older than six months showed a 5.6-mm improvement in CVA and a 4.6-mm improvement in CVA, respectively [22]. Xia et al. reported that helmet therapy is useful for infants older than six months with mild or moderate craniofacial asymmetry [22]. In 2011, Kluba et al. reported that it is necessary to begin helmet therapy before 5–6 months of age for infants with positional plagiocephaly [23]. Aihara et al. found satisfactory results for infants between 4 and 6 months of age [24]. In 2012, Yong Oock Kim et al. found poor outcomes for infants who began treatment after age nine months with a treatment duration of less than eight months [15].

In this study, we divided patients who began treatment between three and five months of age (12–23 weeks group) and after six months of age (≥24 weeks group). Treatment effects were not significantly different between the groups (*p* = 0.345). This may have been due to the inclusion of infants aged six, seven, and eight months, for whom the therapy has been confirmed to be effective, which led to the lack of a significant difference when compared with infants between the ages of three and five months. However, treatment duration was significantly longer in the ≥24 weeks group than in the 12–23 weeks group (*p* < 0.05). Therefore, helmet therapy seems to lead to positive outcomes for infants between three and five months, slow changes for infants older than eight months, and variable outcomes for infants six or seven months of age.

An infant’s head varies in shape and grows quickly, and continuously placing an infant in a supine position induces positional plagiocephaly, increasing its prevalence. Therefore, helmet therapy produces better results when started early, between three and five months of age, than after age eight months.

In 2017, Kreutz et al. reported that facial asymmetry was effectively corrected through helmet therapy in 71 infants with severe deformational skull base plagiocephaly [25]. While studying the causes of varying effects of treatment (even in infants of similar ages), we hypothesized that the size of the anterior fontanelle would have a role. This is because the anterior fontanelle increases in size for the first two months of life and then decreases thereafter [10]; therefore, it can be considered an index of cranial growth during the prenatal period and the period after birth. This growth pattern may affect the usefulness of helmet therapy. In 2019, Wendling-Keim et al. reported that for patients with a small anterior fontanelle and, therefore, lower remolding potential, helmet treatment was more effective than physiotherapy [26].

In the entire study population, changes in the CVAI, PSR, and OSR were larger in infants with larger anterior fontanelles (Figure 3 and Figure 4, Table 3). Furthermore, because the anterior fontanelle size decreases with age, we divided five-month-old infants based on their anterior fontanelle size to perform a more accurate analysis of treatment effects. Similar results were found for groups I, II, and III (Table 4) [15].

Treatment effects may vary depending on the severity of plagiocephaly. Group X (CVA < 10) and group Y (CVA ≥10) were divided into two groups depending on the severity of the plagiocephaly. The CVAI differences according to the anterior fontanel size were statistically significantly correlated in both groups (Table 5).

Changes in the PSR and OSR were greater than changes in the ASR after helmet therapy, suggesting that helmet therapy is more effective for posterior deformation than for anterior skull deformation. Other reports have suggested that helmet therapy is ineffective. In 2010, Hutchison et al. published a study comparing positional therapy and helmet therapy for 126 infants with plagiocephaly [27]. They reported that there were no differences in the correction of plagiocephaly between the helmet-therapy and no-helmet-therapy groups. In 2014, Wijk et al. studied 84 infants and reported that there were no differences in the prognosis of skull deformation between the natural course and helmet-therapy groups [28].

Despite some disagreement, most studies reported that helmet therapy is effective for improving and preventing the progression of plagiocephaly. It is crucial to predict the outcomes of helmet therapy so they can be communicated to guardians. We found that patients with greater fontanelle sizes appeared to demonstrate larger cranial bone movement. Therefore, the anterior fontanelle size could act as a prognostic factor for estimating outcomes.

There were some limitations to this study. The size of the anterior fontanelle was determined using an anterior radiograph; therefore, it was difficult to establish the exact anterior fontanelle size. Additionally, there might have been some bias related to the small number of patients compared to the entire patient population.

## 5. Conclusions

This study found that helmet therapy produces better outcomes in infants with larger anterior fontanelles. Helmet therapy is more effective for correcting posterior skull deformation than anterior skull deformation, and it is more effective when the treatment is begun between the ages of three and five months. Better outcomes of helmet therapy during the earlier months of life may have occurred because younger infants have larger anterior fontanelles.

## Figures and Tables

**Figure 1 jcm-08-01977-f001:**
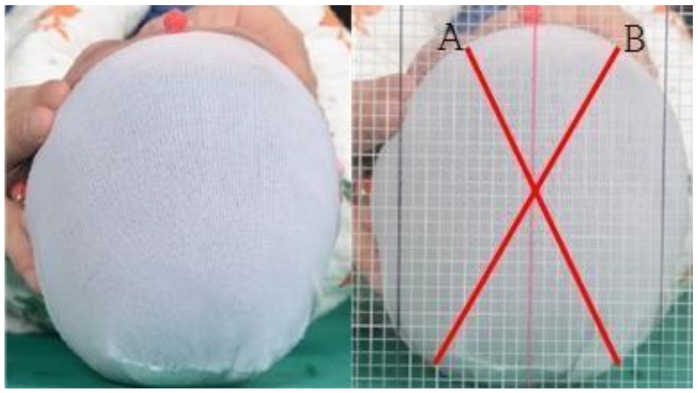
Cranial vault asymmetry (CVA) and CVA index (CVAI) measurements. CVA = |A-B|(cm), CVAI = CVA/A or B (smaller one) × 100(%).

**Figure 2 jcm-08-01977-f002:**
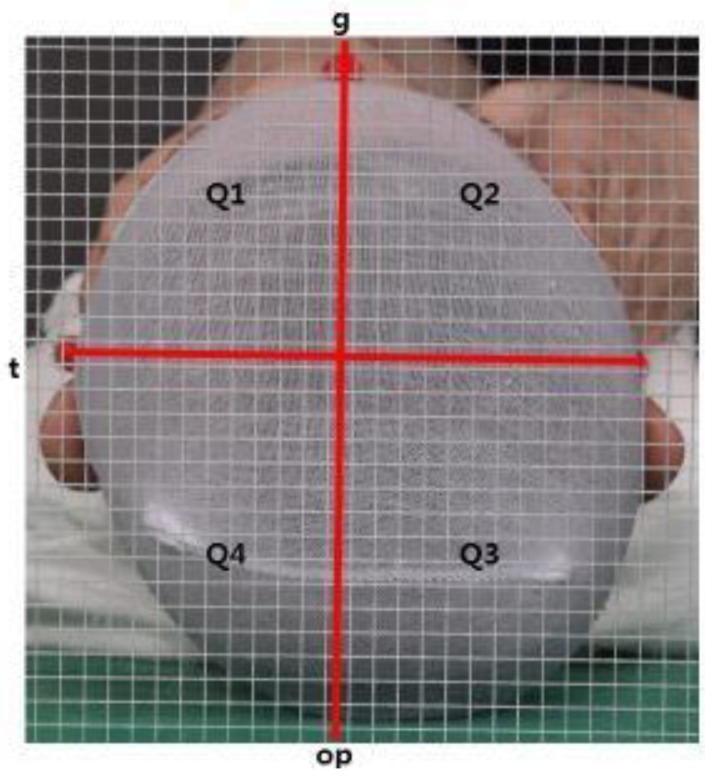
Anterior symmetry ratio (ASR), posterior symmetry ratio (PSR), and overall symmetry ratio (OSR) measurements. ASR = Q1, Q2 smaller one/Q1, Q2 bigger one PSR = Q3, Q4 smaller one/Q3, Q4 bigger one OSR = Q1 + Q4, Q2 + Q3 smaller one/Q1 + Q4, Q2 + Q3 bigger one.

**Figure 3 jcm-08-01977-f003:**
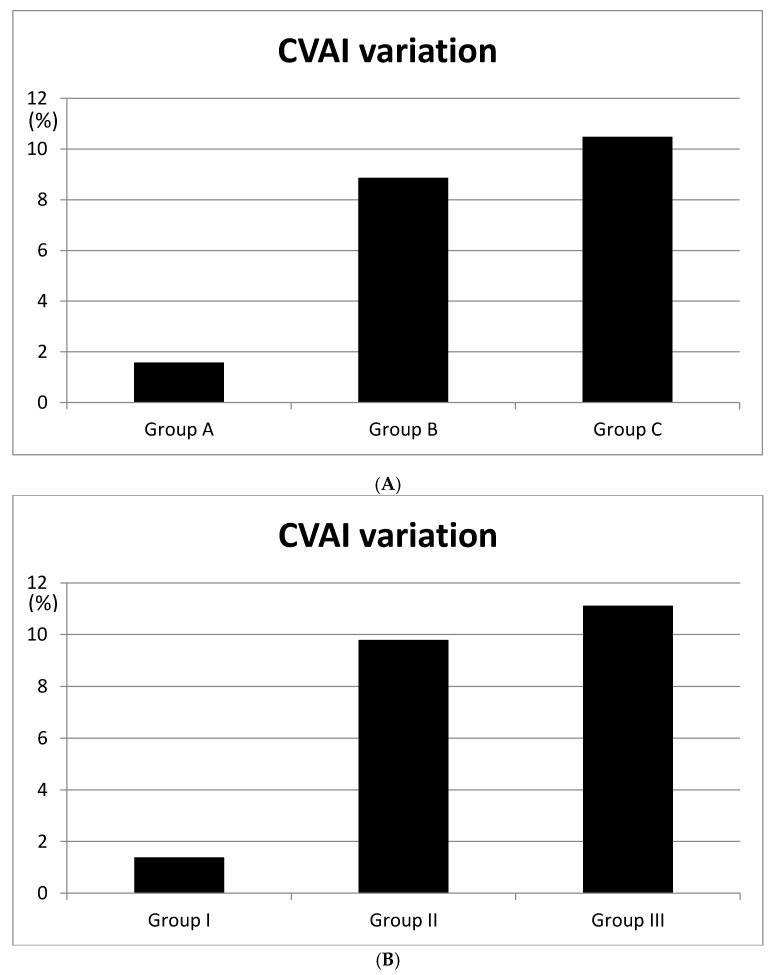
Cranial vault asymmetry index (CVAI) variation according to the anterior fontanelle size. (**A**) CVAI variation were checked 1.58% in Group A, 8.86% in Group B, 10.48% in Group C. (**B**) CVAI variation were checked 1.38% in Group I, 9.79% in Group II, 11.12% in Group III.

**Figure 4 jcm-08-01977-f004:**
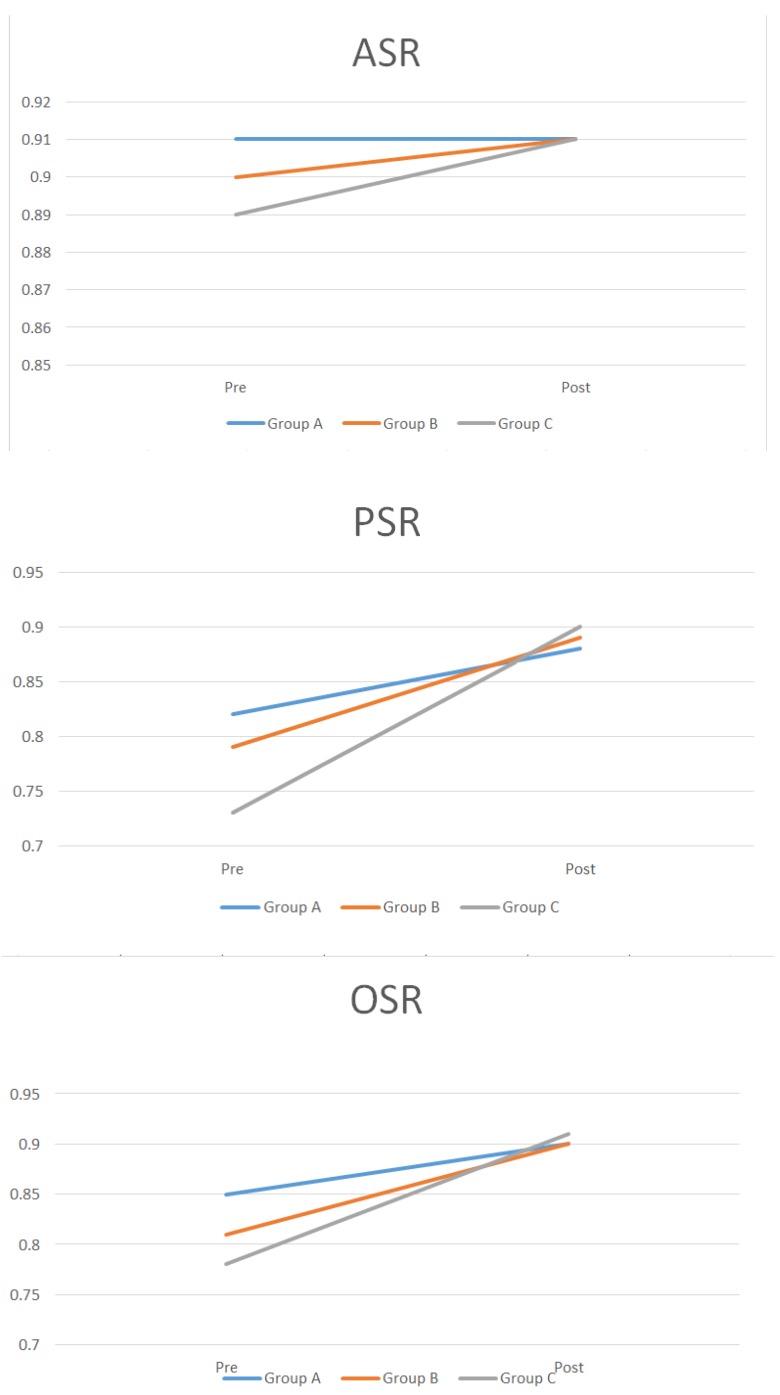
Anterior symmetry ratio (ASR), posterior symmetry ratio (PSR), and overall symmetry ratio (OSR) variations according to the anterior fontanelle size.

**Table 1 jcm-08-01977-t001:** Effects of helmet therapy according to age at initiation.

	12 wk–23 wk (*n* = 152)			24 wk– (*n* = 48)		
	Pre	Post	Duration	Pre	Post	Duration
CVA (mm)	11.4	7.36	12.5 wk	11.11	7.6	16.3 wk * (*p* < 0.05)
CVAI (%)	8.73	5.4		8.5	5.5	

CVA: cranial vault asymmetry (mm); CVAI: cranial vault asymmetry index (%). * statistical significance.

**Table 2 jcm-08-01977-t002:** Patient demographics.

Patient Demographics		
**Total**		200
	M	105
	F	95
Age	12 wk–23 wk	152
	24 wk–27 wk	35
	28 wk–	13
Grouped by anterior fontanelle size		
	A (0–25%)	53
	B (25–75%)	102
	C (75–100%)	45
Grouped by anterior fontanelle size (20wk–23wk infants)		
	I (0–25%)	21
	II (25–75%)	41
	III (75–100%)	20
Grouped by treatment initiation		
	12 wk–23 wk	152
	24 wk–	48

**Table 3 jcm-08-01977-t003:** Effects of helmet therapy according to the anterior fontanelle size.

	All		Group A		Group B		Group C	
	Pre	Post	Pre	Post	Pre	Post	Pre	Post
CVA (mm)	11.4	5.36	7.5	6.23	13.21	5.14	15.4	4.63
CVAI (%)	12.73	4.91	6.5	4.92	13.81	4.95	15.31	4.83
ASR	0.9	0.91	0.91	0.91	0. 9	0.91	0.89	0.91
PSR	0.78	0.88	0.82	0.88	0.79	0.89	0.73	0.9
OSR	0.83	0.92	0.85	0.9	0.81	0.9	0.78	0.91

CVA: cranial vault asymmetry; CVAI: cranial vault asymmetry index; PSR: posterior symmetry ratio; OSR: overall symmetry ratio.

**Table 4 jcm-08-01977-t004:** Effects of helmet therapy according to the anterior fontanelle size at five months (20–23 weeks).

	All		Group I		Group II		Group III	
	Pre	Post	Pre	Post	Pre	Post	Pre	Post
CVA (mm)	11.4	5.36	6.5	6.21	14.15	5.66	16.49	6.1
CVAI (%)	12.73	4.91	6.3	4.92	13.81	4.02	15.81	4.69
ASR	0.9	0.91	0.91	0.91	0.9	0.91	0.9	0.91
PSR	0.78	0.88	0.84	0.86	0.82	0.88	0.78	0.91
OSR	0.83	0.92	0.88	0.91	0.86	0.92	0.86	0.93

CVA: cranial vault asymmetry; CVAI: cranial vault asymmetry index; PSR: posterior symmetry ratio.

**Table 5 jcm-08-01977-t005:** Linear regression test of the change in the CVAI value before and after helmet treatment according to anterior fontanelle size in group X (CVA < 10) and group Y (CVA ≥ 10).

Group X	Group Y
b = 2.073 + 0.12 × a (*p* = 0.003) *	b = 4.184 + 0.09 × a (*p* = 0.006) *
* a = anterior fontanelle size (mm), b = CVAI difference (%)	

* statistical significance.

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
