# Peer review of "The Effects of Helmet Therapy Relative to the Size of the Anterior Fontanelle in Nonsynostotic Plagiocephaly: A Retrospective Study"

_jcm, 2019, doi:10.3390/jcm8111977_

Round 1

Reviewer 1 Report

I have read this paper with great interest, and have provided my comments consecutively as observed in the paper, but overall we need more information on the methods to better understand what you are reporting on. 

Please check the title: lagiocephaly should read plagiocephaly;

Where there any preterms involved (since dichotomous on ? postnatal age)

In the current abstract, it is unclear how the ‘size’ of the anterior fontanelle has been quantified and measured ?

To what extent is there a correlation between fontanel size vs the other assessment tools commonly used to quatify plagiocephaly.

Was the intervention limited to helmet therapy, or where cases also treated with eg physiotherapy ?

Methods section:

in its current version, it is not clear to the reviewer and the reader where the 200 cases are based all, all included ? all data available, a flow chart should be considered, in order to ensure that there are no biases, or if so, that the readership can assess this.  

What is the dichotomous approach on postnatal age based on, and what these age categories as cut off values ? have the authors considered analysis based on continuous variables ?

Is this correct that an X ray is standard practice in your center ? does this mean that the anterio-posterior length of the fontanel has been measured, and not eg cm2 ? Based on the last alinea of the discussion, it may rather be the transverse length (since anterior rx is suggested).

It is not clear how the 5 month data are used: does this mean that x-ray have been collected during treatment, so that this does not serve as an ‘a priori’ predictor but a covariate of ongoing treatment ?

Table 1 needs for sure additional editing.

I may have missed this, but what are the absolute values for the anterior fontanele size

(and check writing, fontanelle or fontanele ?)

Table 3 in my reading does not provide anterior fontanelle size ?

Results and table 5 related comment: why not a continuous assessment, instead of another a priori dichotomous approach.

What were the stopping rules to stop helmet intervention.

The figures should be considered on their relevance, and should provide the information on the X and Y axis in at least the legend (mm, cm, other)

Author Response

Please check the title: lagiocephaly should read plagiocephaly;

Ans : correct word

Where there any preterms involved (since dichotomous on ? postnatal age)

Ans : We did not include children under 32 weeks of premature birth in the study.

And we state about relation postnatal age and anterior fontanelle size at line 71 - 74.

We did not include preterm infants under 32 weeks in the study, G.Duc et al. reported that the gestational age does not show any difference in anterior fontanelle size after full term. Therefore, it is believed that there is no difference according to the gestational age, since all infants who performed the helmet therapy in our paper were targeted only after full term.[37]

In the current abstract, it is unclear how the ‘size’ of the anterior fontanelle has been quantified and measured ?

Ans : Line 77-79

We measured the anterior fontanelle size using X-ray imaging. The transverse length of the anterior fontanelle was measured in the skull AP view conducted to check craniosynostosis before treatment.

To what extent is there a correlation between fontanel size vs the other assessment tools commonly used to quatify plagiocephaly.

Ans : We took your advice.

In each group, we conducted a linear regression test to find a relationship with the size of anterior fontanele by CVA, CVAI, AAR, PAR, and OAR. Within each group, the anterior fontanele size and each of the values was checked to have a positive correlation but not be statistically significance. The change in CVAI, PAR in groups B and C were statistically significant. Statistical significance of CVAI variation is described in other tables. We thought CVA, CVAI was the most appropriate tool to measure the degree of plagiocephaly. Because AAR, PAR, and OAR are figures comparing symmetry by the area of the head of both sides, it is not considered appropriate to reflect the plagiocephaly.

Was the intervention limited to helmet therapy, or where cases also treated with eg physiotherapy ?

Ans : We performed intervention for under two months infants, and helmet therapy for those who were not corrected for more than three months after physical therapy.

Methods section:

in its current version, it is not clear to the reviewer and the reader where the 200 cases are based all, all included ? all data available, a flow chart should be considered, in order to ensure that there are no biases, or if so, that the readership can assess this.  

Ans : We have included all 200 cases. The study was a chart review retrospective study, many outpatient come our clinic for a plagiocephaly assessment, and some patients do not need helmet therapy. We conducted a study on 200 infants whose helmet therapy ended, in order to analyze the association between helmet therapy and anterior fontanelle size.

What is the dichotomous approach on postnatal age based on, and what these age categories as cut off values ? have the authors considered analysis based on continuous variables ?

Ans: Several studies in the Reference have shown that the helmet therapy under six months is more useful than more, and we also divided two groups to assess the effectiveness of the helmet therapy. In addition, the findings were divided into 2 groups for easy clinical explanation of patients' care givers.

Is this correct that an X ray is standard practice in your center ? does this mean that the anterio-posterior length of the fontanel has been measured, and not eg cm2 ? Based on the last alinea of the discussion, it may rather be the transverse length (since anterior rx is suggested).

Ans : We perform x-rays at the start of treatment in all patients who perform helmet therapy to check craniosynostosis. AP and lateral view were implemented, but the length between ant. and post. could not be accurately measured. Therefore, anterior fontanelle size was measured using transverse length, which can be the limitation this paper.

It is not clear how the 5 month data are used: does this mean that x-ray have been collected during treatment, so that this does not serve as an ‘a priori’ predictor but a covariate of ongoing treatment ?

Ans : The age at the beginning of treatment is 5 months. Anterior fontanelle size get smaller in size as time goes by. In order to solve the bias for the treatment effect of helmet therapy with age, the relationship between fontanelle size and helmet therapy was checked only for infants who are 5 months old at the beginning of treatment.

Table 1 needs for sure additional editing.

Ans :

I may have missed this, but what are the absolute values for the anterior fontanele size

(and check writing, fontanelle or fontanele ?)

Ans : Anterior fontelle size was divided based on transverse length. According to length, 0-25%, 25–75% and 75-100% groups were divided. 

Table 3 in my reading does not provide anterior fontanelle size ?

Ans : Anterior fontanelle size is presented as being divided into three groups.

Results and table 5 related comment: why not a continuous assessment, instead of another a priori dichotomous approach.

Ans : In the first review, the other reviewer's advice was taken and Table 5.

Attached is the advice and answer of the previous reviewer.

Tables 3 and 5 show that the groups with smaller anterior fontanelles generally had

more symmetric shapes to begin treatment, conversely, the groups with larger

fontanelles had more starting asymmetry. If the orthotic goal was to achieve a

certain asymmetry ratio (appears to be above 90% based on your tables), then the

infants with worse asymmetries would have longer treatment times and therefore

have greater overall improvement if the same growth rate was observed across the

groups. You do mention that older ages took longer in treatment. I am concerned

that by having groups with statistically different starting values, you would introduce

a treatment bias in your reporting. Could you perhaps present an initial severitymatched

sub group with differing anterior fontanelle sizes (regardless of age)

then examine which group had greater correction or greater treatment times if

there is no difference in correction?Answer) We took your advice and divided it into 2 groups according to the degree

of plagiocephaly and conducted a linear reform test. Content was added to the

abstract and to the text.

--Abstract : Line 20 – 23

Treatment effects of helmet therapy were also analyzed according to the

severity of plagiocephaly in each group as the change in cranial vault

asymmetry index (CVAI) according to the size of the anterior fontanelle.

--Materials and methods : Line 216-225

In addition, to verify the effectiveness of helmet therapy according to the

severity of the plagiocephaly, the CVA values were divided into group X (n=105),

with a CVA value of less than 10, and group Y (n=95), with a CVA value of ≥10

at the beginning of the treatment. In each group, the CVAI difference according

to the anterior fontanelle size was measured before and after helmet treatment.

To identify the correlation between the CVAI difference before and after helmet

therapy according to the anterior fontanelle size, a linear regression analysis

was conducted. Group X showed a positive correlation between the size of the

anterior fontanelle and the CVAI difference that was statistically significant

(p=0.006). In addition, group Y showed a positive correlation between the size

of the anterior fontanelle and the CVAI difference that was statistically

significant (p=0.003) (Table 5).

--Result : Add Table 5.

Table 5. Linear regression test of the change in CVAI value before and after helmet treatment

according to anterior fontanelle size in Group X(CVA 10 or less, Group Y (CVA 10 more).

What were the stopping rules to stop helmet intervention.

Ans: Helmet therapy begins three months after birth, and the duration of helmet therapy varies depending on the degree. Generally, if the OSR is 90% or greater, stop helmet therapy.

The figures should be considered on their relevance, and should provide the information on the X and Y axis in at least the legend (mm, cm, other)

Ans : all figure and table check and adds the legend.

---- Thank you for your advice.

Reviewer 2 Report

Thank you for addressing all of my suggestions and including some new information and clarifications!  I am excited to see this work published as I believe this to be clinically relevant information.  I have a few very minor edits/suggestions prior to final publication:

1) line 47-49: "Furthermore, unilateral... and facial asymmetry.", please reference this statement.

2) I believe there are a few areas where the font may have changed or the paragraph indentations (ex: line 71 seems to either have 2 font sizes or 2 different fonts; line 87-88 seems to have different paragraph formatting; line 265 - 267 seems o also have different line/paragraph formatting).  I imagine this might be corrected in the final edits by the JCM.

3) Please clarify lines 77-78 when you discuss presentation as "mean +/- SD", you explain later how you divided the A, B, and C, I'm wondering if the "mean +/- SD" is a type-o since I do not see information presented in this way.

4) In the abstract, please also spell out "cranial vault asymmetry index" before you abbreviate it to CVAI.

5) I know you explain it in the text, but it would be helpful to also list your ASR and PSR Formulas under Figure 2 like you do for CVA and CVAI in Figure 1.

6) It may just be my PDF copy, but the columns on Table 1 do not appear to line up properly.

7) Table 2 is missing the number of infants in groups I, II, and III.

8) Under Table 5, please have either "*statistically significant" or "*statistical significance" instead of "*statistically significance"

9) Lines 263-265 are nearly identical to 54-55.  Please either use the same reference [12] or state that your study supports the results of Kim et al.

I appreciate all your efforts on this study!

Author Response

1) line 47-49: "Furthermore, unilateral... and facial asymmetry.", please reference this statement.

Ans : add reference

Il Yung Moon, So Young Lim, Kap Sung Oh, Analysis of Facial Asymmetry in Deformational Plagiocephaly Using Three Dimensional Computed Tomographic Review. Arch Craniofac Surg. 2014 Dec; 15(3): 109-116 Captier G, Leboucq N, Bigorre M, Canovas F, Bonnel F, Bonnafe A,Montoya P. Plagiocephaly: morphometry of skull base asymmetry. Surg Radiol Anat 2003;25:226-33 Kelly KM, Littlefield TR, Pomatto JK, Ripley CE, Beals SP, Joganic EF. Importance of early recognition and treatment of deformational plagiocephaly with orthotic cranioplasty. Cleft Palate Craniofac J 1999;36:127-30 St John D, Mulliken JB, Kaban LB, Padwa BL. Anthropometric analysis of mandibular asymmetry in infants with deformational posterior plagiocephaly. J Oral Maxillofac Surg 2002;60:873-7

2) I believe there are a few areas where the font may have changed or the paragraph indentations (ex: line 71 seems to either have 2 font sizes or 2 different fonts; line 87-88 seems to have different paragraph formatting; line 265 - 267 seems o also have different line/paragraph formatting).  I imagine this might be corrected in the final edits by the JCM.

Ans : correct font and paragraph identification

3) Please clarify lines 77-78 when you discuss presentation as "mean +/- SD", you explain later how you divided the A, B, and C, I'm wondering if the "mean +/- SD" is a type-o since I do not see information presented in this way.

Ans : (34.3mm ± 19.1 [SD]; group A: 0-25%; group B: 25-75%; group C: 75-100%).

4) In the abstract, please also spell out "cranial vault asymmetry index" before you abbreviate it to CVAI.

Ans : at line 22, writing cranial vault asymmetry inedex (CVAI) and after At line 23, cranial vault asymmetry (CVA), CVAI

5) I know you explain it in the text, but it would be helpful to also list your ASR and PSR Formulas under Figure 2 like you do for CVA and CVAI in Figure 1.

Ans :

Figure 2. Anterior symmetry ratio (ASR), posterior symmetry ratio (PSR), and overall symmetry ratio (OSR) measurements.

ASR = Q1, Q2 smaller one / Q1, Q2 bigger one

PSR = Q3, Q4 smaller one / Q3, Q4 bigger one

OSR =Q1 + Q4, Q2 + Q3 smaller one / Q1 + Q4, Q2 + Q3 bigger one

6) It may just be my PDF copy, but the columns on Table 1 do not appear to line up properly.

Ans : I’m sorry, correct it.

Before correct

After correct

7) Table 2 is missing the number of infants in groups I, II, and III.

Ans : correct

8) Under Table 5, please have either "*statistically significant" or "*statistical significance" instead of "*statistically significance"

Ans : correct

9) Lines 263-265 are nearly identical to 54-55.  Please either use the same reference [12] or state that your study supports the results of Kim et al.

Ans : Correct

Furthermore, because the anterior fontanelle size decreases with age, we divided 5-month-old infants according to their anterior fontanelle size to perform a more accurate analysis of treatment effects. Similar results were found for groups I, II, and III (Table 4) [12].

Thank you for your advice.

Round 2

Reviewer 1 Report

thank you for the revisions, i still have two questions left and one comment

comment: what you have measured is fontanel length, not so much size, if i get this right (since only one length as assessed by x ray)

questions

it is still not clear how the 200 cases were recruited, included, and if there were exclusion. I recommend to add a flow chart with such information

finally, you have subdivided your group by centile length values, but can you please also add the absolute values ? to make your data useful for comparison 

Author Response

thank you for the revisions, i still have two questions left and one comment

comment: what you have measured is fontanel length, not so much size, if i get this right (since only one length as assessed by x ray)

Ans : Thank you for your good comments.

questions

it is still not clear how the 200 cases were recruited, included, and if there were exclusion. I recommend to add a flow chart with such information

Ans : Line 70-73

Using x-ray, we diagnosed positional plagiocephaly infants without cranio synostosis. We explained the usefulness of helmet therapy to the caregiver, treatment was performed if the caregiver wanted a helmet therapy.

From 1 January 2016, we recruited 200 nonsynostotic plagiocephaly infants from the 1st of January 2016 to enroll the progress and observable infants through clinical photography after the helmet therapy was conducted.

finally, you have subdivided your group by centile length values, but can you please also add the absolute values ? to make your data useful for comparison 

Ans : Line 82-86 correct

group A: 0–25%(0 - 5.1mm); group B: 25–75%(5.1 – 8.7mm); group C: 75–100%(8.7mm-). In addition, 5-month-old infants were divided into group I (0~25%, 0 - 5.2mm), group II (25~75%, 5.2 – 8.9mm), and group III (75~100%, 8.9mm -) based on their anterior fontanelle size.

Thank you

This manuscript is a resubmission of an earlier submission. The following is a list of the peer review reports and author responses from that submission.

Round 1

Reviewer 1 Report

The authors have put together a wonderful manuscript relating the size of the anterior fontanelle to the amount of correction achieved with helmet therapy.  The inclusion of a single-age subdivision (5 month old infants into groups I, II, and III) is important to report and appreciated.

There are some areas which can be improved to strengthen the paper prior to publication:

1) The introduction needs to include more references.  The authors make strong claims without reference support such as the first paragraph (lines 32-24), the sentences on lines 42-45 ("This is the most common reason ... leads to skull base and facial asymmetry."), and the sentence on lines 53-56 ("This is because the anterior fontanelle increases in size during the first two months of life, decreasing after this period.")  Please reference these statements.

2) In the introduction, you describe a novel method for cranial measurement - is this referring to measuring the size of the anterior fontanelle via x-rays? (Line 51), if so, please clarify the novel method.  Do you have a previously published study which you can reference in this paragraph?

3) In the Materials & Methods section, the authors split the groups into <25%, 25-75%, and >75% groups, but there is no explanation of how to calculate this percentage.  Please define this.

4) Please check table formatting on Table 1 - the columns do not line up the column titles.

5) Tables 3 and 5 show that the groups with smaller anterior fontanelles generally had more symmetric shapes to begin treatment, conversely, the groups with larger fontanelles had more starting asymmetry.  If the orthotic goal was to achieve a certain asymmetry ratio (appears to be above 90% based on your tables), then the infants with worse asymmetries would have longer treatment times and therefore have greater overall improvement if the same growth rate was observed across the groups.  You do mention that older ages took longer in treatment.  I am concerned that by having groups with statistically different starting values, you would introduce a treatment bias in your reporting.  Could you perhaps present an initial severity-matched sub group with differing  anterior fontanelle sizes (regardless of age) then examine which group had greater correction or greater treatment times if there is no difference in correction? 

6) Did all of your 5 month old infants have the same or similar gestational age? Generally, premature infants grow at a faster rate than full-term infants because they are functionally younger.

7) The titles or captions of Figure 3 and Figure 4 would benefit from more detail to clarify what is being presented.

8) Line 192 of your conclusion states you found correction in facial asymmetry - this was not reported in the results.  Please report this in your results or eliminate this paragraph.  OSR, PSR, and ASR show improvements in the neurocranium with helmet use, but these ratios do not report on facial asymmetry.

9) I believe line 83, just before Figure 1 may be missing a less than symbol - proposed: "normal <3.5%"

10) I was able to find a recently published article of a related study, please include it in either your introduction or discussion:

Childs Nerv Syst. 2019 Jun 17. doi: 10.1007/s00381-019-04215-y. [Epub ahead of print]

A new parameter for the management of positional plagiocephaly: the size of the anterior fontanelle matters.

Wendling-Keim DS1, Macé Y2, Lochbihler H3, Dietz HG2, Lehner M2,4.

Overall, I am excited to see this study and it has very good information to report.  I want to thank the authors for examining the anterior fontanelle size in relationship to helmet correction.  I have seen this correlation clinically, but until seeing this article and that of Wendling-Keim's group, I had not seen it published previously.